The mitochondrial genome of Paragonimus westermani (Kerbert, 1878), the Indian isolate of the lung fluke representative of the family Paragonimidae (Trematoda)

Biswal Devendra K. 1
Chatterjee Anupam 2
Bhattacharya Alok 3 alok.bhattacharya@gmail.com
Tandon Veena 1 4 tandonveena@gmail.com
1 Bioinformatics Centre, North-Eastern Hill University , Shillong, Meghalaya , India
2 Department of Biotechnology and Bioinformatics, North-Eastern Hill University , Shillong, Meghalaya , India
3 School of Life Sciences, Jawaharlal Nehru University , New Delhi , India
4 Department of Zoology, North-Eastern Hill University , Shillong, Meghalaya , India
Nakai Kenta
Electronic publication date: 2014 Aug 12
Publication date: 2014
Volume: 2
Electronic Location ID: e484
Received 2014 May 25; Accepted 2014 Jun 23
Copyright: © 2014 Biswal et al.
Copyright year: 2014
Copyright holder: Biswal et al.
License: This is an open access article distributed under the terms of the Creative Commons Attribution License, which permits unrestricted use, distribution, reproduction and adaptation in any medium and for any purpose provided that it is properly attributed. For attribution, the original author(s), title, publication source (PeerJ) and either DOI or URL of the article must be cited.
License URL: https://creativecommons.org/licenses/by/4.0/

Keywords: Paragonimus, Next generation sequencing, Mitochondria, Bayesian analysis, Transfer RNA

Funding: Department of Biotechnology, Govt. of India BT/48/NE/TBP/2010 Indian Council of Medical Research Project The work was supported by the Department of Biotechnology, Government of India under the DBT-NER Twinning program sanctioned to VT, AB and DKB and partly by the Indian Council of Medical Research Project on worm zoonoses sanctioned to VT (Principal Investogator). The funders had no role in study design, data collection and analysis, decision to publish, or preparation of the manuscript.

==============================
Among helminth parasites, Paragonimus (zoonotic lung fluke) gains considerable importance from veterinary and medical points of view because of its diversified effect on its host. Nearly fifty species of Paragonimus have been described across the globe. It is estimated that more than 20 million people are infected worldwide and the best known species is Paragonimus westermani, whose type locality is probably India and which infects millions of people in Asia causing disease symptoms that mimic tuberculosis. Human infections occur through eating raw crustaceans containing metacercarie or ingestion of uncooked meat of paratenic hosts such as pigs. Though the fluke is known to parasitize a wide range of mammalian hosts representing as many as eleven families, the status of its prevalence, host range, pathogenic manifestations and its possible survivors in nature from where the human beings contract the infection is not well documented in India. We took advantage of the whole genome sequence data for P. westermani, generated by Next Generation Sequencing, and its comparison with the existing data for the P. westermani for comparative mt DNA phylogenomic analyses. Specific primers were designed for the 12 protein coding genes with the aid of existing P. westermani mtDNA as the reference. The Ion torrent next generation sequencing platform was harnessed to completely sequence the mitochondrial genome, and applied innovative approaches to bioinformatically assemble and annotate it. A strategic PCR primer design utilizing the whole genome sequence data from P. westermani enabled us to design specific primers capable of amplifying all regions of the mitochondrial genome from P. westermani. Assembly of NGS data from libraries enriched in mtDNA sequence by PCR gave rise to a total of 11 contigs spanning the entire 14.7 kb mt DNA sequence of P. westermani available at NCBI. We conducted gap-filling by traditional Sanger sequencing to fill in the gaps. Annotation of non-protein coding genes successfully identified tRNA regions for the 24 tRNAs coded in mtDNA and 12 protein coding genes. Bayesian phylogenetic analyses of the concatenated protein coding genes placed P. westermani within the family Opisthorchida. The complete mtDNA sequence of P. westermani is 15,004 base pairs long; the lung fluke is the major etiological agent of paragonimiasis and the first Indian representative for the family Paragonimidae to be fully sequenced that provides important genetic markers for ecological, population and biogeographical studies and molecular diagnostic of digeneans that cause trematodiases.

Introduction

Among about 50 known species of the genus Paragonimus, Paragonimus westermani, one of the causative agents of paragonimiasis, was first described as early as 1878 and is the most well-known species within the genus Paragonimus because of its wide geographical distribution and medical importance (Blair, Xu & Agatsuma, 1999). Typically, paragonimiasis is a disease of the lungs and pleural cavity but extra-pulmonary paragonimiasis also happens to be an important clinical manifestation. It is a neglected disease that has received poor attention from public health authorities. As per the recent estimates, about 293 million people are at risk, while several millions are infected worldwide (Keiser & Utzinger, 2009). However, this may be an underestimate as there are still many places where the disease burden has yet to be assessed. There has been an increased recognition of the public health importance of paragonimiasis and other foodborne trematodiases in recent times (Fried, Graczyk & Tamang, 2004) and some serious concern for Paragonimus species outside endemic areas owing to the risk of infection through food habits in today’s globalized food supply. In the case of paragonimiasis, this resurgence of interest can partly be attributed to the common diagnostic confusion of paragonimiasis with tuberculosis, as symptoms of the former closely mimic those of the latter, thereby leading to an inappropriate treatment being administered especially in areas where both tuberculosis and paragonimiasis co-occur and create overlapping health issues (Toscano et al., 1995). The state-of-the-art molecular biology techniques, next generation sequencing (NGS) technology and their rapid development in contemporary times may provide additional tools for the differential identification of digenean trematode infections to overcome limitations of current morphology-based diagnostic methods. Owing to their high nucleotide substitution rates, parasitic flatworm mitochondrial (mt) genomes have become very popular markers for diagnostic purposes and for resolving their phylogenetic relationships at different taxonomic ranks. Comparative mitochondrial genomics can provide more reliable results and reveal important informations of mtDNA architectural features such as gene order and structure of non-coding regions.

At present there have been reports on two isolates of P. westermani mtDNA, one diploid (2n) mtDNA (incomplete) from Leyte Island, Philippines that resembles P. westermani morphologically and is sometimes regarded as a subspecies, P. westermani filipinus (Sato et al., 2003) and one triploid (3n) complete Korean P. westermani isolate mtDNA (accession: NC_002354 complete, unpublished). In our present study, we determined the complete mtDNA nucleotide sequence of P. westermani, which was collected from several sites in Changlang District, Arunachal Pradesh in India, using NGS data generated from total genomic DNA extracts. Phylogenetic analyses were carried out using a supermatrix of all the concatenated mt sequences of 12 protein-coding genes of digenean trematode and cestodes, (taking nematode species as an outgroup) available in public domain (GenBank). This newly sequenced Indian isolate P. westermani mt genome sequence along with the one in the RefseQ database bearing accession NC_002354 of NCBI would provide useful information on both genomics and Paragonimidae evolution, including the biogeographic status of the cryptic species of the lung flukes and other mtDNA sequences available for any member of the trematode group.

Methods

Parasite material and DNA extraction

Naturally infected freshwater edible crabs (Barytelphusa lugubris lugubris) were collected from Changlang District in Arunachal Pradesh (altitude—213 mASL, longitude—96°15′N and latitude—27°30′E). The isolation of metacercariae from the crustacean host muscle tissues was carried out by digestion technique using artificial gastric juice. The 70% alcohol-fixed metacercariae were further processed for DNA extraction and PCR amplification. The lysed individual worms were subjected to DNA extraction by standard ethanol precipitation technique (Sambrook, Fitsch & Maniatis, 1989); DNA was also extracted from individual metacercarie on FTA cards with the aid of Whatman’s FTA Purification Reagent.

Primer design strategy and PCR

Illumina reads from our unpublished P. westermani whole genome data were mapped to P. westermani reference sequence (gi|23957831| ref| NC_002354.2 |). The alignment was carried out using Bowtie aligner. The mapped reads were extracted in fastq format using custom perl script. We obtained 62,874 paired end reads, which aligned to different intervals in the P. westermani mt genome, covering ∼3 kb of the 15 kb mt genome (NC_002354.2). Accordingly, primers were designed at these regions, using sequence information from reference to ensure optimum primer design (File S1). We conducted PCR using 10 ng of genomic DNA from P. westermani with the following PCR conditions: 10 ng of FD-2 DNA with 10 µM Primer mix in 10 µl reaction, PCR thermo cycling conditions – 98 °C for 3 min, 35 cycles of 98 °C for 30 s, 60 °C for 30 s, 72 °C for 1 min 30 s, final extension 72 °C for 3 min and 4 °C hold. We gel-eluted the bands (File S1) corresponding to different products, pooled these products and proceeded to NGS library construction. These clean single end reads were also further used for bioinformatics analysis in this study. The Illumina mito mapped reads were quality checked using proprietary tool SeqQC (Genotypic Technology Pvt. Ltd., Bangalore, India). The QC reads are outlined in Table 5.

NGS library construction, sequencing and assembly

DNA was subjected to a series of enzymatic reactions that repair frayed ends, phosphorylate the fragments, and add a single nucleotide ‘A’ overhang and ligate adaptors (Illumina’s TruSeq DNA sample preparation kit). Sample cleanup was done using Ampure XP SPRI beads. After ligation, ∼300–350 bp fragment for short insert libraries and ∼500–550 bp fragment for long insert libraries were size-selected by gel electrophoresis, gel extracted and purified using Minelute columns (Qiagen). The libraries were amplified using 10 cycles of PCR for enrichment of adapter-ligated fragments. The prepared libraries were quantified using Nanodrop and validated for quality by running an aliquot on High Sensitivity Bioanalyzer Chip (Agilent). 2X KapaHiFiHotstart PCR ready mix (Kapa Biosystems Inc., Woburn, MA) reagent was used for PCR. The Ion Torrent library was made using Ion Plus Fragment library preparation kit (Life Technologies, Carlsbad, US) and the Illumina library was constructed using TruSeqTM DNA Sample Preparation Kit (Illumina, Inc., US) reagents for library prep and TruSeq PE Cluster kit v2 along withTruSeq SBS kit v5 36 cycle sequencing kit (Illumina, Inc., US) for sequencing (Biswal et al., 2013). PCR products were sonicated, adapter ligated and amplified for x cycles to generate a library and subsequently were sequenced to generate reads of an average of 121 nt SE reads on Ion Torrent. The IonTorrent raw data was processed for 3′ low quality bases trimming, and adapter contamination. Since the Ion Torrent data might have had host contamination, the processed reads were then aligned to the reference sequence of Paragonimus westermani mtDNA (NC_002354) available in GenBank, Department of Environmental Health Science, Kochi Medical School, Oko, Nankoku, Kochi, Japan. The alignment was carried out using Tmap Ion Torrent proprietary tool. The mapped reads were extracted in fastq format using custom perl script. These clean reads were used for further bioinformatics analysis in this study. The processed reads as well as mito-mapped reads were quality checked using proprietary tool SeqQC (Genotypic Technology Pvt. Ltd., Bangalore, India).

De-novo assembly

The Ion Torrent-mapped reads were assembled using Newbler (Quinn et al., 2008) software. The Illumina-mapped reads were subjected to reference assisted de novo assembly using velvet (Zerbino & Birney, 2008) assembler. Quite a few hash lengths were tested for velvetg. Hash length 65 gave the optimal results in terms of total contig length, N50, and maximum contig length. Therefore, k-mer 65 assembly was considered for further analysis. Sanger reads were also added in the final assembly. The draft sequence was generated using Ion Torrent reads, Illumina reads, Sanger reads, hybrid high-quality de novo assembly and subsequently the de novo-leftout regions were obtained using reference assisted assembly and consensus calling. Extensive manual curation work was carried out to produce the complete sequence. The complete sequence comprises 15,004 bases in total. There were a few regions in the mitochondrial sequence, namely ∼900 bases in the start and ∼1,500 bases in the end, where there were few or no sequences at 3x depth. In that case, the consensus sequence was retrieved using VCFtools (Danecek et al., 2011). The consensus sequence was introduced at such regions; the sequences in question are represented by letters in lower case of nucleotides, while the confident regions are represented in upper case in the fasta sequence file.

In silico analysis for nucleotide sequence statistics, protein coding genes (PCGs) prediction, annotation and tRNA prediction

Sequences were assembled and edited by using CLC Genome Workbench V.6.02 with comparison to published flatworm genomes and the assembled whole single mtDNA contig was annotated with the aid of ORF finder tool at NCBI (http://www.ncbi.nlm.nih.gov/gorf/gorf.html) and MITOS, which were subsequently used to search for homologous digenean trematode PCGs already housed in REFSEQ NCBI database (http://www.ncbi.nlm.nih.gov/refseq/) by using tBLASTn (Altschul et al., 1990). The program ARWEN (Laslett & Canbäck, 2008) was used to identify the tRNA genes by setting the search to predict secondary structures occasionally with very low Cove scores (<0.5) and, where necessary, also by restricting searches to find tRNAs lacking DHU arms (using the trematode tRNA option). Nucleotide codon usage for each protein-encoding gene was predicted using the program Codon Usage at (http://www.bioinformatics.org/sms2/codon_usage.html).The ORFs and codon usage profiles of PCGs were analyzed. The newly sequenced and assembled P. westermani mtDNA was annotated using MITOS and the output file was further used to sketch the newly sequenced genome with GenomeVX at http://wolfe.ucd.ie/GenomeVx/.

Phylogenetic analysis

DNA sequences of the 12 protein-coding genes from 13 representative trematode, cestode and nematode species were retrieved (Table 1), aligned in clustal w and concatenated using MESQUITE (Maddison & Maddison, 2011). The supermatrix was used for generating phylogenetic trees using Bayesian analysis in MrBayes v3.1 (Ronquist & Huelsenbeck, 2003). The mt genome sequences of the nematode Ascaris suum and Ascaris lumbricoides were used as an outgroup. For the nucleotide alignment, the GTR + I + G model was used and Bayesian analysis was run for 1,000,000 generations and sampled every 1,000 generations. The first 25% of trees were omitted as burn-in and the remaining trees were used to calculate Bayesian posterior probabilities retaining the trees with a majority consensus rule of 50.

Table 1 mt DNA nucleotide sequence statistics information of representative helminth parasites.

Sequence
type	DNA	DNA	DNA	DNA	DNA	DNA	DNA	DNA	DNA	DNA	DNA	DNA	DNA	
Length	14,118 bp
circular	14,462 bp
circular	15,004 bp
circular	14,277 bp
circular	14,014 bp
circular	13,875 bp
circular	14,478 bp
circular	14,415 bp
circular	14,085 bp
circular	13,670 bp
circular	13,709 bp
circular	14,281 bp
circular	14,284 bp
circular	
Organism
Name	Fasciolopsis
buski	Fasciola
hepatica	Paragonimus
westermani	Opisthorchis
felineus	Paramphistomum
cervi	Clonorchis
sinensis	Fasciola
gigantica	Schistosoma
mansoni	Schistosoma
japonicum	Taenia
saginata	Taenia
solium	Ascaris
lumbricoides	Ascaris
suum	
Accession	Submitted to
GenBank	NC_002546	NC_002354	EU921260	NC_023095	FJ381664	NC_024025	NC_002545	NC_002544	NC_009938	NC_004022	JN801161	NC_001327	
Modification
Date	submitted	01-FEB-2010	submitted	18-AUG-2010	14-JAN-2014	01-JUL-2010	01-MAY-2014	14-APR-2009	01-FEB-2010	14-APR-2009	01-FEB-2010	01-DEC-2011	11-MAR-2010	
Weight
(single-stranded)	4,396.507
kDa	4,499.496
kDa	4,666.455
kDa	4,437.683
kDa	4,363.551
kDa	4,311.834
kDa	4,504.913
kDa	4,482.165
kDa	4,371.002
kDa	4,242.425
kDa	4,251.992
kDa	4,428.619
kDa	4,429.981
kDa	
Weight
(double-stranded)	8,721.667
kDa	8,934.244
kDa	9,270.244
kDa	8,820.283
kDa	8,657.348
kDa	8,571.888
kDa	8,944.06
kDa	8,904.302
kDa	8,700.11
kDa	8,443.711
kDa	8,467.723
kDa	8,443.711
kDa	8,822.899
kDa	
Annotation table	
Feature
type	Count	Count	Count	Count	Count	Count	Count	Count	Count	Count	Count	Count	Count	
CDS	12	12	12	12	12	12	12	12	12	12	12	12	12	
Gene	12	12	12	12	12	12	12	12	12	12	12	12	12	
Misc.
feature	1	1	–	–	–	–		1	1	–	–	1	2	
rRNA	2	2	2	2	2	2	2	2	2	2	2	2	2	
tRNA	22	22	24	22	22	22	22	23	23	22	22	22	22	

Results & Discussion

Mitochondrial genome organisation of P. westermani mtDNA

The two rRNA genes and 12 protein coding genes, typical of the flatworms, were identfied by comparison of their sequence similarity and secondary structures with those of other flatworms. The mt genome lacks ATP synthase protein 8 (ATP8) with no characteristic amino acid signatures. Over a long time gene order remains stable in animal mtDNAs (Boore, 1999; Saccone et al., 1999). Differences in the mtDNA gene order between members of the same family, though rare, can occur in higher taxonomic ranks. A marked difference in the gene order was found among the various trematode, cestode and nematode species as outlined in Fig. 1. The total length for the digenean P. westermani (AF219379) is 14,965 bp, and for Schistosoma japonicum (NC_002544) and S. mansoni (NC_002545) is approximately 14.5 kb as curated by the NCBI staff. Other digeneans possess small mt genomes. The mtDNA sequence of P. westermani (Bioproject accession number PRJNA248332, Biosample accession sample SAMN02797822 and SRA SRX550161) is 15,004 bp in length and is well within the range of typical metazoan mtDNA sizes (14–18 kb). The mt genome of P. westermani is larger than that of other digenean species available in GenBank (http://www.ncbi.nlm.nih.gov/genbank/) to date (Table 1). It contains 12 protein-coding genes (cox1-3, nad1-6, nad4L, atp6 and cytb), 24 transfer RNA (tRNA) genes and 2 ribosomal RNA genes (rrnL and rrnS) (Fig. 2 and Table 2). The gene arrangement pact of protein-coding genes in P. westermani tallies with that of Fasciola hepatica (Le et al., 2000; Le, Blair & McManus, 2001), Opisthorchis felineus (Shekhovtsov et al., 2010), Fasciola gigantica (Liu et al., 2014), Fasciolopsis buski (Biswal et al., 2013) and Paramphistomum cervi (Yan et al., 2013) mt genomes, but is different from that seen in Taenia and Ascaris species (Nakao, Sako & Ito, 2003; Okimoto, Macfarlane & Wolstenholme, 1990) (Fig. 3). An overlapping region spanning nearly 40 bp between 3′ nad4L end and nad4 5′ end was also seen in P. westermani, a feature common to other digenean trematodes. The 12 protein coding genes and their blast hit protein plots are summarised in Fig. 4. The protein plot shows for each gene and each position the quality value if it is above the threshold; the different genes are differentiated with a range of colour codes. Basically, the initial hits used in MITOS (Bernt et al., 2013) correspond to the “mountains” in this plot that visualizes the signal from the BLAST searches (Altschul et al., 1990). The arrows shown on the top of the plot depict the gene order annotation and the quality values are shown on a log scale.

Figure 1 Comparative Synteny map of the representative species for the helminth mtDNA illustrating the protein coding genes, tRNAs, rRNAs etc.

Figure 2 Circular genome map of Paragonimus westermani mtDNA.

The manual and in-silico annotations with appropriate regions for P. westermani mtDNA and annotated GenBank flat file for P. westermani were drawn into a circular graph in GenomeVX depicting the 12 PCGs and 24tRNAs.

Figure 3 Inferred Phylogenetic relationship among the representative helminth mtDNA species of the concatenated 12 protein coding genes.

Trees were inferred using MrBayes v3.1. Posterior support values are given at nodes. Differences in the gene order in the mitochondrial genomes of parasitic flatworms from the Trematoda and Cestoda and taking Nematoda (Ascaridida) as an outgroup are indicated on the phylogenetic leaf nodes. See text for more details.

Figure 4 Summarized 12 protein coding genes and their blast hit protein plots.

The protein plot depicts the quality value for each gene and each position if it is above the threshold and the different genes are differentiated with a range of colour codes. The hits used in MITOS correspond to the “mountains” in this protein plot that visualizes the signal from the BLAST searches. The arrows shown on the top of the plot depict gene order annotation and the quality values are shown on a log scale.

Table 2 P. westermani mtDNA annotations showing PCGs and tRNA in dot bracket format.

Name	Start	Stop	Strand	Length	Structure	
cox3	658	1134	+	477		
trnH(gtg)	1147	1209	+	63	(((((((..((((......)))).(((((.......)))))....((.(..).))))))))).	
cob-a	1213	1860	+	648		
cob-b	1922	2311	+	390		
nad4l	2393	2644	+	252		
nad4_0-a	2922	3167	+	246		
nad4_0-b	3163	3465	+	303		
nad4_1-b	3564	3710	−	147		
nad4_1-a	3725	3805	−	81		
trnQ(—)	3882	3944	+	63	(((((((..((((......)))).(((((......)))))....((.......))))))))).	
trnF(gaa)	3951	4020	+	70	((((.((..((((........)))).(((((.......)))))....(((.........))))).)))).	
trnM(cat)	4027	4092	+	66	(((((((..((((........)))).(((((.......)))))....((((...))))))))))).	
atp6	4326	4583	+	258		
nad2	4627	5262	+	636		
trnV(tac)	5470	5531	+	62	(((((.(..((((.....)))).(((((.......)))))....(((....)))).))))).	
trnA(tgc)	5539	5610	+	72	(((((((..((((............)))).(((((.......)))))....(((((...)))))))))))).	
trnD(gtc)	5615	5681	+	67	(((((((..((((.........)))).(((((.......)))))....(((.....)))))))))).	
nad1-a	5767	5877	+	111		
nad1-b	6077	6535	+	459		
trnN(gtt)	6606	6675	+	70	(((((((..((((........)))).(((((.......)))))....(((((.....)))))))))))).	
trnP(tgg)	6676	6743	+	68	((((.(((..((((.......)))).(((((.......)))))....(((.......)))))))))).	
trnI(gat)	6749	6812	+	64	((((((.(..((((....)))).(((((.......)))))....(((((..)))))))))))).	
trnK(ctt)	6815	6880	+	66	(((((((..((((......)))).(((((.......)))))....(((((...)))))))))))).	
nad3	7001	7231	+	231		
trnS1(gct)	7244	7302	+	59	(((((((.......(((((.......)))))....(((((......)))))))))))).	
trnW(tca)	7308	7375	+	68	(((((((..((((......)))).(((((.......)))))....((((.......))))))))))).	
cox1	7379	8872	+	1494		
trnT(tgt)	8914	8977	+	64	((((((...((((.......)))).(((((.......)))))....(((....))).)))))).	
rrnL	9067	9181	+	115	((...((((((((.....((.(((((((.((((...))))...
((..(((((.....)))))..))..).)))))).))....((.....)).......))))))))......))	
rrnL	9417	9951	+	535	.................(((....)))....(.....)..................((................))..
((....))..........................................((...................(...)......
(((........))).....))..........((((((........))))))...... (((.(((.((.....
((((((((((((.(((((...)))))..((((...)))).((.............)).........
((((...)))).....)))))..))))))).......((((((((((((...
((.......))...))))))))).))).........(((((.((((((..(((.((((((......)))))).)))..
(((((.....)))))....)))))..)...))))).....(((((((....))).))))....))..))))))....
((...............))...........	
trnC(gca)	9961	10025	+	65	(((((((..((((...))))...(((((.......)))))....(((((...)))))))))))).	
rrnS	10028	10751	+	724	...(((((.......))))).(((((((...((((((((....((..(....)....... (((..................
(((..((...)).))).))))).....(((.(..(((((....))))))))).))))))))...
((((((((..................)))...............))))))))))))....((((.....((((.(.
(((..........(((...((......))...)))............((....))...))).).))))....(((((...
((((.........))))...))))).............)))).((((....))))........((((.
((((((((.......(((((..((((((((((....(((........))).........(((((((.....((.((..
((((((((((....(((.....(....).
((....)).)))..............))).)))...))))))))....))))))).)).)))))))).............
(((((..........)))))..........))))).....
((((((...........))))))...........)))))))))).))..............((.
(((....))).))................(((((((.((....)).)))))))...........	
cox2-a	11020	11094	+	75		
cox2-b	11112	11204	+	93		
cox2-c	11201	11338	+	138		
nad6	11410	11772	+	363		
trnY(gta)	11814	11876	+	63	.((((((..((((.......)))).(((((.......)))))....((((.))))))))))..	
trnL1(tag)	11883	11947	+	65	.((((((..(((.......))).(((((.......)))))....(((.(...).)))))))))..	
trnL2(—)	12025	12086	+	62	(((((((..(((.......))).((((........))))....(((((.)))))))))))).	
trnR(tcg)	12091	12154	+	64	(((((((((......)))).(((((.......)))))....(((((.......)))))))))).	
nad5_0-a	12430	12927	+	498		
nad5_0-b	12989	13285	+	297		
nad5_1	13506	13733	+	228		
trnG(tcc)	13751	13820	+	70	(((((((..((((..........)))).(((.(.......).)))....((((.....))))))))))).	
trnE(ttc)	14358	14422	+	65	(((((((..((((.......)))).(((((.......)))))....((((...))))))))))).	

Comparison of mtDNA between P. westermani of Indian and Korean isolates

The complete P. westermani Indian isolate mtDNA comprises of 15,004 bases in total while the Korean isolate (NC_002354) is of 14,965 bp in length. Out of 15,004 bases in the sequences, 13,188 bases were confident bases (87.88% of total), while 1,818 bases were low quality bases (12.11% of total). Mapping of assembled mitochondria against the reference Korean isolate was carried out using online Blastn that show 85% identical bases between the two, with 99% query coverage with the best possible e-value of 0.0 and with a maximum score of 12,579. A dot plot matrix view was generated depicting the sequence similarity regions on the reference sequence. The x-axis represents the assembled sequence, whereas y-axis represents the reference sequence (Fig. 5A). In order to generate visual output of the mapped assembled mtDNA against the reference mtDNA, standalone blast and Artemis Comparison Tool (ACT) was incorporated (Carver et al., 2005). Sequence similarity map (Fig. 5B) shows dark red links where high % identical synteny is found between reference and query sequence. No complete NR is known for P. westermani in both the Indian and Korean mtDNA. The melting temperatures, count and frequency of atoms in both single stranded and double stranded DNA, count and frequency of nucleotides showed little variation and are outlined in Table 4. The percentage nucleotide variation for A and T was higher in Indian isolate compared to the Korean mtDNA while the G, C percentage was higher in the Korean isolate (Fig. 6). In both the mtDNAs there are 12 protein coding genes and 1 rRNA with a variation in the number of tRNAs i.e., 24 in the Indian isolate as compared to the Korean mtDNA with 23 tRNAs.

Figure 5 Dot plot matrix and sequence similarity map depicting the the sequence similarity regions between the assembled and reference mtDNA.

(A) Dot plot matrix between the reference and assembled mt DNA. X-axis represents the assembled sequence, whereas y-axis represents the reference sequence. (B) Visual output of the mapped assembled mtDNA against the reference mtDNA using standalone blast and Artemis Comparison Tool (ACT).

Figure 6 Comparative histogram of the nucleotide frequences of the Indian and Korean P. westermani isolates.

Blue coloured bars indicate Indian isolate while dark coloured bars indicate the reference Korean isolate deposited in GenBank.

Genetic code, nucleotide composition and codon usage

It is a well established fact that mtDNA of parasitic flatworms uses AAA to specify ASN (Lys in the universal code), AGA and AGG to specify Ser (Arg in the universal code), and TGA to specify Trp (stop codon in the universal code). ATG is the usual start codon while GTG and other codons are also used as start codons (Le, Blair & McManus, 2002). The P. westermani mtDNA exhibited ATG and ATA as start codons and TAG and TAA as stop codons (Table 3). mtDNA genomes of invertebrates have a tendency to be AT-rich (Wolstenholme, 1992), a feature common in several parasitic flatworm protein coding genes. However, the nucleotide composition is not uniform among the species. For Schistosoma mansoni, the AT-rich percentage is 68.7%, whereas for Fasciola hepatica it is 63.5% AT and for P. westermani only 54.6% AT (Le, Blair & McManus, 2002). The nucleotide composition in the P. westermani Indian isolate was biased towards G and T, which is similar to that of other digeneans, viz. F. hepatica, O. felineus, C. sinensis, P. cervi; unlike S. japonicum and other schistosomes, which are more biased towards A and T. The atomic composition in single stranded DNA exhibits hydrogen with a frequency of 37.5%, carbon 29.8%, nitrogen 10.8%, oxygen 18.8% and phosphorus 3.0% (Table 4).

Table 3 Codon usage for Paragonimus westermani mt DNA.

AmAcid	Codon	Number	/1000	Fraction	
Ala	GCG	57.00	11.40	0.27	
Ala	GCA	38.00	7.60	0.18	
Ala	GCT	75.00	15.00	0.36	
Ala	GCC	39.00	7.80	0.19	
Cys	TGT	208.00	41.59	0.76	
Cys	TGC	67.00	13.40	0.24	
Asp	GAT	91.00	18.20	0.72	
Asp	GAC	36.00	7.20	0.28	
Glu	GAG	111.00	22.20	0.69	
Glu	GAA	51.00	10.20	0.31	
Phe	TTT	310.00	61.99	0.74	
Phe	TTC	109.00	21.80	0.26	
Gly	GGG	168.00	33.59	0.34	
Gly	GGA	89.00	17.80	0.18	
Gly	GGT	166.00	33.19	0.34	
Gly	GGC	66.00	13.20	0.13	
His	CAT	44.00	8.80	0.61	
His	CAC	28.00	5.60	0.39	
Ile	ATT	97.00	19.40	0.71	
Ile	ATC	40.00	8.00	0.29	
Lys	AAG	66.00	13.20	1.00	
Leu	TTG	226.00	45.19	0.34	
Leu	TTA	110.00	22.00	0.17	
Leu	CTG	92.00	18.40	0.14	
Leu	CTA	35.00	7.00	0.05	
Leu	CTT	147.00	29.39	0.22	
Leu	CTC	56.00	11.20	0.08	
Met	ATG	89.00	17.80	0.80	
Met	ATA	22.00	4.40	0.20	
Asn	AAA	55.00	11.00	0.45	
Asn	AAT	44.00	8.80	0.36	
Asn	AAC	24.00	4.80	0.20	
Pro	CCG	34.00	6.80	0.26	
Pro	CCA	20.00	4.00	0.15	
Pro	CCT	57.00	11.40	0.43	
Pro	CCC	22.00	4.40	0.17	
Gln	CAG	42.00	8.40	0.64	
Gln	CAA	24.00	4.80	0.36	
Arg	CGG	51.00	10.20	0.35	
Arg	CGA	26.00	5.20	0.18	
Arg	CGT	51.00	10.20	0.35	
Arg	CGC	19.00	3.80	0.13	
Ser	AGG	125.00	25.00	0.21	
Ser	AGA	57.00	11.40	0.09	
Ser	AGT	76.00	15.20	0.13	
Ser	AGC	36.00	7.20	0.06	
Ser	TCG	56.00	11.20	0.09	
Ser	TCA	52.00	10.40	0.09	
Ser	TCT	134.00	26.79	0.22	
Ser	TCC	68.00	13.60	0.11	
Thr	ACG	37.00	7.40	0.29	
Thr	ACA	20.00	4.00	0.16	
Thr	ACT	43.00	8.60	0.34	
Thr	ACC	27.00	5.40	0.21	
Val	GTG	156.00	31.19	0.29	
Val	GTA	58.00	11.60	0.11	
Val	GTT	256.00	51.19	0.48	
Val	GTC	65.00	13.00	0.12	
Trp	TGG	159.00	31.79	0.58	
Trp	TGA	113.00	22.60	0.42	
Tyr	TAT	74.00	14.80	0.57	
Tyr	TAC	55.00	11.00	0.43	
End	TAG	66.00	13.20	0.50	
End	TAA	66.00	13.20	0.50	

Table 4 Comparative nucleotide sequence statistics of mtDNA between P. westermani Indian and Korean isolates.

	P. westermani Indian	P. westermani Korean (NC_002354)	
Sequence information	
Information			
Sequence type	DNA	DNA	
Length	15,004	14,965	
Weight (single-stranded)	4666.455	4652.101	
Weight (double-stranded)	9270.244	9246.535	
Melting temperatures—degrees celsius	
[salt]			
0.1	83.53	84.71	
0.2	88.53	89.7	
0.3	91.45	92.63	
0.4	93.53	94.7	
0.5	95.14	96.31	
Counts of annotations	
Feature type			
CDS	12	12	
Gene	12	12	
Source	1	1	
rRNA	1	1	
tRNA	24	23	
Counts of atoms (As single-stranded)	
Ambiguous residues are omitted in atom counts.	
Atoms			
Hydrogen (H)	185,664	184,951	
Carbon (C)	147,756	147,080	
Nitrogen (N)	53,610	53,530	
Oxygen (O)	93,068	92,834	
Phosphorus (P)	15,004	14,963	
Counts of atoms (As double-stranded)	
Ambiguous residues are omitted in atom counts.	
Atoms			
Hydrogen (H)	368,285	366,846	
Carbon (C)	293,261	292,031	
Nitrogen (N)	111,847	111,970	
Oxygen (O)	180,050	179,556	
Phosphorus (P)	30,008	29,926	
Frequencies of atoms	
As single-stranded	
Ambiguous residues are omitted in atom counts.	
Atoms			
Hydrogen (H)	0.375	0.375	
Carbon (C)	0.298	0.298	
Nitrogen (N)	0.108	0.109	
Oxygen (O)	0.188	0.188	
Phosphorus (P)	0.03	0.03	
As double-stranded	
Ambiguous residues are omitted in atom counts.	
Atoms			
Hydrogen (H)	0.374	0.374	
Carbon (C)	0.298	0.298	
Nitrogen (N)	0.114	0.114	
Oxygen (O)	0.183	0.183	
Phosphorus (P)	0.031	0.031	
Counts of nucleotides	
Nucleotide			
Adenine (A)	2571	2339	
Cytosine (C)	2284	2550	
Guanine (G)	4535	4679	
Thymine (T)	5614	5395	
Any nucleotide (N)	0	2	
C + G	6819	7229	
A + T	8185	7734	
Frequencies of nucleotides	
Nucleotide			
Adenine (A)	0.171	0.156	
Cytosine (C)	0.152	0.17	
Guanine (G)	0.302	0.313	
Thymine (T)	0.374	0.361	
Any nucleotide (N)	0	0	
C + G	0.454	0.483	
A + T	0.546	0.517	

Table 5 Summary of illumina and Ion-Torrent quality control reads.

Ion torrent reads	
S.No	1	2	
Fastq file name	processed_reads.fastq	mapped_mito.fastq	
Fastq file size	239.71 MB	71.55 MB	
Time taken for analysis	8.75 s	2.76 s	
Maximum read length	260	260	
Minimum read length	35	35	
Mean Read Length	121	117	
Total number of reads	890,504	292,832	
Total number of HQ reads 1*	890,442	292,822	
Percentage of HQ reads	99.993%	99.997%	
Total number of bases	107,866,584 bases	34,145,801 bases	
Total number of bases in Mb	107.8666 Mb	34.1458 Mb	
Total number of HQ bases 2*	105,216,008 bases	33,218,357 bases	
Total number of HQ bases in Mb	105.2160 Mb	33.2184 Mb	
Percentage of HQ bases	97.543%	97.284%	
Total number of non-ATGC characters	0 bases	0 bases	
Total number of non-ATGC characters in Mb	0.000000 Mb	0.000000 Mb	
Percentage of non-ATGC characters	0.000%	0.000%	
Number of reads with non-ATGC characters	0	0	
Percentage of reads with non-ATGC characters	0.000%	0.000%	
Illumina reads	
S.No	1	
Fastq file name	SE_ill.fastq	
Fastq file size	14.56 MB	
Time taken for analysis	0.48 s	
Maximum read length	100	
Minimum read length	50	
Mean read length	96	
Total number of reads	62,874	
Total number of HQ reads 1*	62,874	
Percentage of HQ reads	100.000%	
Total number of bases	6,053,872 bases	
Total number of bases in Mb	6.0539 Mb	
Total number of HQ bases 2*	5,982,733 bases	
Total number of HQ bases in Mb	5.9827 Mb	
Percentage of HQ bases	98.825%	
Total number of non-ATGC characters	410 bases	
Total number of non-ATGC characters in Mb	0.000410 Mb	
Percentage of non-ATGC characters	0.007%	
Number of reads with non-ATGC characters	240	
Percentage of reads with non-ATGC characters	0.382%	

Transfer and ribosomal RNA genes section

A standard cloverleaf structure is generally seen for most of the tRNAs. There are exceptions that include tRNA(S), in which the paired dihydrouridine (DHU) arm is missing as in all parasitic flatworm species and tRNA(A), in which the paired DHU-arm is missing as in cestodes contrary to trematodes. Previous studies indicate structures for tRNA(C) that somewhat vary among the parasitic flatworms. In some species, a paired DHU-arm is missing (Schistosoma mekongi and cestodes), whereas it is present in others (F. hepatica and F. buski). It is noteworthy that the P. westermani Indian isolate exhibited 24 tRNA genes, 1 TV replacement loop tRNA genes and 2 D replacement loop tRNA genes. The tRNA GC range varied from 37.9% to 59.4% (Fig. 7). Ribosomal large and small subunits in parasitic flatworms are unremarkable. They are smaller than those in most other metazoans but can be folded into a recognizable, conserved secondary structures (Le, Blair & McManus, 2001). The rrnL (16S ribosomal RNA) and rrnS (12S ribosomal RNA) genes of P. westermani were identified by sequence comparison with those of cloesly related trematodes and these ribosomal genes were separated by tRNA-C (GCA).

Figure 7 24 tRNA secondary structures predicted using ARWEN.

Non-coding regions

There are one or two longer non-coding region(s) (NR) in every genome comprising stable stem–loop structures that are associated with genome replication or repeat sequences. Previous studies report repeats in the NR of many animal mt genomes that may be an outcome of slippage-mismatching mechanisms (Le, Blair & McManus, 2001). In parasitic flatworms, NRs vary in length and complexity. The NR is divided by one or more tRNA genes into a SNR and a LNR in digenean trematodes. A common feature of LNRs is the presence of long repeats. In the present study the P. westermani mtDNA though didn’t exhibit significant demarcation of LNR and SNR, there were regions with repeats with total number of 3,158 variants with a total of 1,722 SNPs and 1,436 INDELS.

Phylogenetic analysis

Several genetic markers from nuclear rDNA regions and mtDNA of flukes have been employed in some systematic and population genetic studies of helminth parasites. As of now the full-length mt genomes of 14 digenean, 34 cestode and 70 nematode species have been determined, characterized, and are published in GenBank. It is confirmed that alignments with more than 10,000 nucleotides from mtDNAs can provide ample information for phylogenetic resolution, hypothesis building and evolutionary interpretation of the major lineages of tapeworms. Use of complete mtDNA sequences for phylogenetic analyses are more reliable and informative (Waeschenbach, Webster & Littlewood, 2012). In the present study, a phylogenetic tree inferred from concatenated nucleotide sequences of the 12 protein-coding genes (shown in Fig. 2) is well supported by very high posterior probabilities (100%). Two large clades are visibly informative: one contains members of the Family Schistosomatidae, and the other includes members representing the sequence of families in order of increasingly derived status: Opisthorchiidae, Paragonimidae, Paramphistomidae and Fasciolidae (Trematoda); Ascarididae (Nematoda) and Taeniidae (Cestoda). This arrangement was seen in the tree based on nucleotide sequences, in which a clade containing Fasciolidae and Paragonimidae members was strongly supported and P. cervi was sister to this clade. P. westermani claded with Opisthorchis felineus and Clonorchis sinensis. Members representing Taeniidae served as an outgroup (Fig. 3).

Conclusions

In this study, we took advantage of the whole genome sequence data generated by NGS technology for P. westermani Indian isolate and its comparison to existing data for the P. westermani (Korean isolate) mitochondrial genome for the purpose of comparative analysis between the mt genomes of the two isolates. Precise and specific primers were designed for amplification of mitochondrial genome sequences from the parasite DNA sample with the help of existing P. westermani mtDNA available in the NCBI Refseq database. Here we present and discuss the complete sequence of the coding region of the mitochondrial genome of P. westermani, the Indian lung fluke isolate, which posesses the same gene order as that of other Digenea (Opisthorchidae and Paramphistomatidae) and consists of 12 PCGs, 24 tRNAs and 2 rRNAs. There are long repetitive regions in the fluke that can serve as diagnostic markers with phylogenetic signals. The complete mtDNA sequence of P. westermani will add to the knowledge of digenean mitochondrial genomics and also provide an important resource for studies of inter- and intra-specific variations, biogeographic studies, heteroplasmy of the flukes belonging to Paragonimidae and a resource for comparative mitochondrial genomics and systematic studies of Digenea in general.

Supplemental Information

File S1 List of primers and gel images of long range PCR product as used in the study for NGS library construction

Click here for additional data file.

We would like to acknowledge Dr. Sudip Ghatani, Department of Zoology, NEHU, Shillong for collecting the biosamples and M/s Genotypic Technologies, Bangalore, India for carrying out NGS sequencing for this project, especially the efforts of Dr. Deepti Saini for the primer design strategy.

Additional Information and Declarations

Competing Interests

Author Contributions

DNA Deposition

The authors declare there are no competing interests.

Devendra K. Biswal conceived and designed the experiments, performed the experiments, analyzed the data, contributed reagents/materials/analysis tools, wrote the paper, prepared figures and/or tables.

Anupam Chatterjee performed the experiments, reviewed drafts of the paper.

Alok Bhattacharya and Veena Tandon conceived and designed the experiments, performed the experiments, analyzed the data, contributed reagents/materials/analysis tools, wrote the paper, reviewed drafts of the paper.

The following information was supplied regarding the deposition of DNA sequences:

Bioproject: PRJNA248332, Biosample: SAMN02797822 and SRX550161.

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
