# Peer review of "The mitochondrial genome of Paragonimus westermani (Kerbert, 1878), the Indian isolate of the lung fluke representative of the family Paragonimidae (Trematoda)"

_PeerJ, doi:10.7717/peerj.484_

## Round 0.1 · original submission · Minor Revisions

Your manuscript has been reviewed by two experts in the field. As you will find, both of them admit the high value of your work but give several important comments that must be addressed before its publication. Please read the comments carefully and revise the manuscript accordingly. Especially, I feel that the apparent contradiction in their phylogenetic tree must be treated more carefully.

·

Basic reporting

The authors analyzed whole genome sequence data for P. westermani, generated by Next Generation Sequencing, and its comparison with the existing data for the P. westermani complete mitochondrial genome sequence to design precise. They are also conducted gap-filling by traditional Sanger sequencing to fill in the gaps, using a strategic PCR primer design for amplifying all regions of the mitochondrial genome from P. westermani. The fully sequenced data in this paper provides important genetic markers. This paper may be published for PeerJ. However, reviewer recommends revising the manuscript with minor comments.

1. The paper is titled “The mitochondrial genome of Paragonimus westermani”. However, it is mentioned in the introduction line 75 that mitochondrial genomes have become very popular markers. It implies that the genome has been completed and used extensively. Therefore, the title (which implies that there has not been any known genome sequence before) seems to contradict line 75. What do the authors want to say in this paper: A completion of a previously incomplete genome sequence using NGS or a truly novel sequence data that has never been published before? The title or the introduction needs revision.
2. Also in the introduction, since NGS is used to complete the mitochondrial genome, then the background of mitochondrial genome should be clear by including the status of the genome. Is it incomplete? How can NGS be used to solve this problem?

Experimental design

1. In the methodology, library preparation should be separated from DNA extraction section.
2. In the methodology, the method describing the obtaining of Illumina reads is quite unclear. The authors state the result of the reads, and then return to PCR technique to obtain the band, and then proceed to NGS library construction. It is should be written in order. If these are two different experiments, then it should be stated here.

Validity of the findings

1. In the conclusions, line 280-281 states, “…for designing precise and specific primers for amplification of mitochondrial genome sequences from the parasite DNA sample.” In my understanding, the authors want to describe the mitochondrial genome of P. westermani and designing the primers is just a method to reach the purpose. Unless the designing of the specific primers is the purpose of the paper, then it is suggested to rephrase the sentence. The statement of the primers is important because it can lead to misunderstanding and non-synchronization with the title. This problem also appears in abstract line 34-35.

Additional comments

1. Line 92-93, metacercariae does need to be capitalized. Also, altitude, longitude, latitude.
2. Line 97, What does “…eggs in FTA card” mean?
3. Line 135-136: 3min → 3 mins, 30sec → 30 secs.
4. Line 142-143: ion torrent → Ion Torrent, illumina → Illumina, denovo → de novo, velvet(Zerbino → velvet (Zerbino.
5. Line 151: 15004 → 15,004.
6. Line 156: 13188 → 13,188.
7. Line 157: 1818 → 1,818.
8. Line 194: flatworms.The → flatworms. The.
9. Line 194: What does atp 8 mean? Atp8?
10. Line 195: longtime → long time.
11. Line 205: TM is not needed.
12. Line 207: (Fig. 2)(Table 2) → (Fig. 2, table 2).
13. Line 231-232: P. cervi and unlike S. japonicum → P. cervi; unlike S. japonicum.
14. Line 233-234: elements do not need to be capitalized.

Reviewer 2 ·

Basic reporting

This paper is a very interesting one in terms of usage of The state-of-the-art molecular biology techniques, next generation sequencing (NGS) technology. However, there is little novelty in the results. Phylogenetic tree constructed seems quite different from trees according to taxonomical standard. For example, the position of Schistosoma in the tree is unusual, and further one of nematoda, Ascaris, should be positioned outside of the cluster of trematodes and cestodes. There are no detailed descriptions about differences between Indian Pw and the existing data for the Pw complete mitochondrial genome sequence. It seems very important to compare between them.

Experimental design

Good

Validity of the findings

There are several important findings but phylogenetic tree results seem not correct.

Additional comments

Need more descriptions between Indian and other Pw.

---

## Round 0.2 · Minor Revisions

I confirm that all of the reviewers' points are properly addressed in the revised manuscript. However, I feel that the new title seems to be a little complicated:

The mitochondrial genome of Paragonimus westermani (Kerbert, 1878) Braun, 1899 the Indian isolate of the lung fluke representative of the family Paragonimidae (Trematoda)

How about a shorter title, such as:
The mitochondrial genome of Paragonimus westermani (Kerbert, 1878),
the Indian isolate of the lung fluke representative of the family Paragonimidae (Trematoda)?

Please note that this is just a suggestion, not the indication from me.

---

## Round 0.3 · accepted · Accept

Thanks for having accepted my suggestion on the title, I feel that this one is much better.